# *Clostridium butyricum* Potentially Improves Immunity and Nutrition through Alteration of the Microbiota and Metabolism of Elderly People with Malnutrition in Long-Term Care

**DOI:** 10.3390/nu14173546

**Published:** 2022-08-28

**Authors:** Lin Liu, Xiang Chen, Lu Liu, Huanlong Qin

**Affiliations:** 1Shanghai Tenth People’s Hospital, Tongji University School of Medicine, Shanghai 200072, China; 2Realbio Genomics Institute, Shanghai 200123, China; 3Pengpu Community Medical Service Center, Shanghai 200436, China

**Keywords:** metagenome, metabolism, probiotic, elderly people

## Abstract

Recent research advances examining the gut microbiome and its association with human health have indicated that microbiota-targeted intervention is a promising means for health modulation. In this study, elderly people in long-term care (aged 83.2 ± 5.3 year) with malnutrition (MNA-SF score ≤ 7) were recruited in a community hospital for a 12-week randomized, single-blind clinical trial with *Clostridium butyricum*. Compared with the basal fluctuations of the control group, an altered gut microbiome was observed in the intervention group, with increased (*p* < 0.05) *Coprobacillus* species, *Carnobacterium divergens*, and *Corynebacterium_massiliense*, and the promoted growth of the beneficial organisms *Akketmanse muciniphila* and *Alistipes putredinis*. A concentrated profile of 14 increased Kyoto Encyclopedia of Genes and Genomes (KEGG) orthologs (KOs) that were enriched in cofactor/vitamin production and carbohydrate metabolism pathways were discovered; the genes were found to be correlated (*p* < 0.05) with an elevated abundance of plasma metabolites and short-chain fatty acids (SCFAs), unsaturated medium- to long-chain fatty acids (MFA, LFA), carnitines, and amino acids, thus suggesting a coordinated ameliorated metabolism. Proinflammatory factor interferon-gamma (IFN-γ) levels decreased (*p* < 0.05) throughout the intervention, while the gut barrier tight junction protein, occludin, rose in abundance (*p* = 0.059), and the sensitive nutrition biomarker prealbumin improved, in contrast to the opposite changes in control. Based on our results obtained during a relatively short intervention time, *C. butyricum* might have great potential for improving nutrition and immunity in elderly people in long-term care with malnutrition through the alteration of gut microbiota, increasing the abundance of beneficial bacteria and activating the metabolism in SCFA and cofactor/vitamin production, bile acid metabolism, along with efficient energy generation.

## 1. Introduction

Aging is a gradual and continuous degenerative process [1]. The compositional structure, diversity richness, and metabolic activities of gut microbiota in elderly people (the aged gut) are versatile and are related to multiple factors, including host characteristics (age, gender, genetics, chronic diseases), intake (diet, drugs), environment (residence care, seasons), and socioeconomic factors [2]. The rapid progress of high-throughput sequencing technologies and multi-omics research, including metagenomics [3], metabolomics [4], and metatranscriptomics, as well as advances in terms of functional, animal, and clinical studies, have promoted our understanding and recognition of gut microbiota as a “super-organ” with an important role in human health. Previous studies have found the gut microbiota to be involved in digestive function, immune response regulation, and neurological modulation through microbial components and products (essential vitamins, SCFAs, secondary bile acids, 5-hydroxytryptamine, acetylcholine, etc.) [5].

Malnutrition, frailty, sarcopenia, and immunosenescence are typical health concerns with high incidence among elderly populations [6]. The aging gut has a relatively low diversity of gut microorganisms, with more individual differences and fewer SCFA-producing species (the core microbes of nine genera: *Phascolarctobacterium, Roseburia, Blautia, Faecalibacterium, Clostridium, Subdoligranulum, Ruminococcus, Coprococcus,* and *Bacteroides* [7,8]). The composition of microbes is also associated with different forms of residential care. Specifically, community-living elderly people have the highest diversity of gut microflora [9] with more *Firmicutes* microorganisms [10], which diversity has been associated with lower frailty, co-morbidity, and inflammation markers, and a better diet (especially the consumption of vegetables and fruits) [11].

It has been found that immunity can be affected by malnutrition through the gut microbiota and gut-associated lymphoid tissue (GALT) [12]; an impaired gut microbiome can hamper gut permeability or cause “gut leakage”, which is associated with susceptibility to pathogenic infections and higher morbidity of multiple chronic diseases including diabetes mellitus, rheumatoid arthritis, cardiovascular disease, constipation, IBD, and cognitive dysfunction, such as depression and dementia [1,13,14]. The close relationship and interaction between the gut microbiota and human health open up the prospect of microbiota-targeted health modulation, such as the supplementation of SCFA-producing probiotics to improve nutrition and immunity [15]. Probiotics constitute an expanding research area and a rapidly progressing field of study; previous studies have reported on their ability to improve malnutrition, neuropsychological disorders [5,16], antibiotic-associated diarrhea (AAD), constipation [16], gut microbiota dysbiosis after surgery, and cancer adjuvant therapy [17,18]. Probiotics benefit the host by the secretion of beneficial metabolites or bacteriocin, to compete against pathogens or promote the growth of beneficial microbes and improve the metabolism and bile acids [19,20,21,22]. With prolonged life expectancy on a global scale and the increased cost of geriatric diseases, the clinical effect of *C. butyricum* on nutrition and immunity improvement in bedridden elderly people suffering from malnutrition is of particular importance; however, so far, limited data are available.

This 12-week, randomized, single-blind clinical trial with *C. butyricum* enrolled a cohort of long-term care subjects in a community hospital (aged 83.2 ± 5.3) suffering from malnutrition (MNA-SF score ≤ 7, average albumin 36.38 ± 2.98 g/L). All the subjects provided informed consent. Initial and final fecal and plasma samples were subjected to quantitative analysis incorporating metagenomics, metabolomics, correlation studies, cytokines (IL-2, IL-4, IL-6, IL-10, IL-17A, and IFN-γ), and immunity biomarkers (occludin, zonulin, CD14, and LBP), as well as nutrition indicators (prealbumin, serum albumin, and hemoglobin). Clinical records and questionnaires were also collected and analyzed by well-trained medical staff (Appendix A). The findings revealed the positive effect of *C. butyricum* on elderly people with malnutrition in relation to multiple aspects of health and promoted our understanding of the effect of probiotic modulation on gut microbiota and health. A further study with a larger sample size and more comprehensive analysis may be needed to thoroughly understand the impact and mechanism of probiotics on the health of elderly people with malnutrition.

## 2. Materials and Methods

### 2.1. Study Design and Enrolled Subjects

Long-term residents in a community hospital in Shanghai, China, were recruited with informed consent for a study of probiotic intervention. The clinical study was approved by the Ethics Committee of Shanghai Tenth People’s Hospital (approved protocol number: 20–161), and registered at the Chinese Clinical Trial Registry (http://www.chictr.org.cn). The recruited participants were randomly divided into a test group (TG) for probiotic intervention with *Clostridium butyricum* (3.5 × 10^5^~3.5 × 10^8^ CFU) after each meal for a continuous period of 12 weeks, alongside a control group (CG) without probiotic intake. The clinical backgrounds of age, gender, BMI, diseases, and nutrition status were comparable in both groups. Initially, 37 persons were randomly assigned to the TG, and 19 to the CG. During the intervention, subjects with the acute onset of diseases, who had left the facility, who had died, who had taken an inconsistent dosage of probiotics, and who took antibiotics or other probiotics such as yogurt (3, 2, 3, 4, and 14 persons, respectively, in the TG; and 1, 0, 1, 2, and 7 persons, respectively, in the CG) were excluded from the later analysis (Figure 1). At the end of the trial, 11 TG participants supplied 25 fecal samples and 22 plasma samples, and 8 CG participants supplied 19 fecal and 16 plasma samples (Table 1 and S1). The samples were subjected to quantitative analyses of metagenomics and metabolomics, as well as immunity and nutrition biomarker examinations indicative of inflammation, gut integrity, and nutrition status. Clinical diagnosis, records, and questionnaires were also collected for intro- and inter-group analysis (Table 1 and S1). The inclusion criteria comprised an age over 65, being at potential risk of malnutrition (MNA-SF score ≤ 7), without serious vital organ disease histories such as heart, lungs, liver, or kidney diseases; the exclusion criteria comprised acute infection, progressive or serious internal diseases, and antibiotic or probiotic history within the previous two months. The demographic data (e.g., age, gender, and BMI), clinical manifestations, observations, and questionnaire records of appetite, defecation routine, and neuropsychic activities of all participants were recorded (Appendix A in the Appendix A). Fecal and blood samples in EDTA were collected at the beginning of the study, after 12 weeks of prebiotic usage, and 12 weeks after probiotic discontinuity. At the end of the intervention, samples from subjects with inconsistent probiotic dosage or antibiotics, who had worsened disease symptoms, who were no longer resident or who had died were excluded from later analysis. Fecal microbial DNA was extracted and subjected to metagenomic analysis. Plasma metabolomes were quantified using the Q300 kit in the UPLC-MS/MS system (for details, please refer to the latter part of the manuscript). Plasma was also used for the quantification of six cytokines indicative of inflammatory activity and immunity by flow cytometry methodology, with four biomarkers of gut permeability tested using ELISA, as well as three proteins representing nutrition status tested with a turbidimetric inhibition immunoassay.

### 2.2. Metagenomic Sequencing, Assembly, and Functional Annotation

Fresh samples collected from the participants were transported to the laboratory with an ice pack within 2 h, aliquoted, and frozen immediately for storage at −80 °C. DNA was extracted from each fecal sample using the QIAamp Fast DNA Stool Mini Kit (Qiagen, Germany) following the manufacturer’s instructions. The quality of the DNA was analyzed using Qubit 2.0 (Invitrogen, Waltham, MA, USA) and 1% agarose gel electrophoresis. The metagenomic DNA libraries were constructed with 2 μg of genomic DNA, according to the TruSeq DNA Sample Prep v2 Guide (Illumina, San Diego, CA, USA), with an average insert size of 500 bp. All libraries were analyzed with an Agilent Bioanalyzer 2100 for quality control (Agilent Technologies, Santa Clara, CA, USA) using a DNA LabChip 1000 kit. Whole-genome shotgun sequencing was conducted on the Illumina HiSeq 4000 platform in paired-end mode, with a read length of 150 bp.

In total, 383 Gb of raw data (6.08 Gb per sample on average, mean insert size 388 ± 19 bp) were generated; low-quality reads and adaptor sequences were filtered from the raw reads of metagenomics sequences. Bases with a quality score of below 30 were trimmed from the 3′ end of the reads, and reads were removed if they were shorter than 70 bp or were mapped to the human genome. Finally, we obtained 379 Gb of clean data (an average of 6.01 Gb per sample), and the average proportion of high-quality reads was approximately 98.04%.

The De Bruijn graph-based assembler SOAP de novo (version 2.04) was employed to assemble short reads, then scaffolds with a minimum length of 500 bp were extracted and broken into gap-free contigs. Open reading frames (ORFs) for each sample were predicted by MetaGene (http://metagene.cb.k.u-tokyo.ac.jp/ (accessed on 15 December 2015)). All sequences of ORFs with a 95% identity and 90% coverage were clustered as the non-redundant gene catalog by the CD-HIT program (version 4.5.7). The final non-redundant gene catalog contained 2,958,198 microbial genes, which had an average length of 686 bp. The functional annotation of genes was carried out via a BLASTP search against the Kyoto Encyclopedia of Genes and Genomes (KEGG) database (e-value ≤ 1 × 10^−5^ and high-scoring segment pair scoring > 60).

### 2.3. Phylogenetic Analysis, Species, and KEGG Ortholog (KO) Abundance Profiling

The clean reads were aligned to the microbial reference genomes collected from the National Center for Biological Information (NCBI, http://www.ncbi.nlm.nih.gov (accessed on Jan 18 2020) by the SOAPalign program (version 2.21). For certain species or genes, reads that were paired-matched to related genomes or genes were split into two parts: (1) U: reads that match a particular genome/gene only, (2) M: reads that also match another genome/gene. The abundance of the species/gene was also split into two parts, Ab(U) and Ab(M). The unique part, Ab(U), was calculated as the number of reads, divided by the length of the genome/gene. For the multiple parts, Ab(M), each of the reads in set M was assigned to several parts, according to the unique abundance of species/gene with which the reads matched [23]. The KO abundance profile was obtained via the accumulation of gene abundance profiling. The abundance of genes belonging to the same KO was accumulated to obtain the value for the abundance of this KO.

### 2.4. Plasma Metabolome Quantification

Fresh whole blood with EDTA was collected from the participants, centrifuged at 4000× *g* for 20 min at 4 °C, then supernatant plasma was aliquoted and frozen immediately for storage at −80 °C. Targeted plasma metabolites were quantified using the Q300 Kit (Metabo-Profile, Shanghai, China), which covers up to 306 metabolites and >12 biochemical classes, utilized according to the instructions of the manufacturer, alongside ultra-performance liquid chromatography coupled with a tandem mass spectrometry (UPLC-MS/MS) system (ACQUITY UPLC-Xevo TQ-S, Waters Corp., Milford, MA, USA).

For data processing, the raw data files generated by UPLC-MS/MS were processed using the MassLynx software (v4.1, Waters, Milford, MA, USA) to perform peak integration, calibration, and quantitation for each metabolite. The self-developed platform, iMAP (v1.0, Metabo-Profile, Shanghai, China), was used for the statistical analyses, including PCA, OPLS-DA, univariate analysis and pathway analysis. The differential metabolites were identified by the Wilcoxon rank-sum test (R package wilcox.test) or Student’s *t*-test (R package *t*-test) with a threshold *p*-value of < 0.05 between the results before (T0) and after (T3) taking probiotics for three months, as well as between the results before (C0) and after three months (C3) for the control group without probiotics.

### 2.5. Differential Metagenomic Analysis

On the basis that a genus or species with low abundance might not properly reflect the actual situation, the genus or species with a median abundance of less than 2 × 10^−8^ in the sample were discarded. The differential genus and the differential species were identified by the Wilcoxon rank-sum test (R package wilcox.test) with a threshold *p*-value < 0.05 between the results before (T0) and after taking probiotics for three months (T3). In the control group without probiotics, the identification of the differential genus and the differential species that changed in the samples before (C0) and after three months (C3) of taking probiotics was the same.

After taking probiotics for three months (T3), the participants then did not take probiotics for the next three months (T6). Based on the hypothesis that the effect of taking probiotics would not get better after discontinuity of probiotics, we filtered the species that were not significant (*p*-value < 0.05) in T3 during the Kruskal–Wallis test (R package kruskal.test) among T0, T3, and T6. The same was the identification of the differential species that changed naturally over time among C0, C3 and C6.

On the basis that the KOs with low abundance might not properly reflect the actual situation, the KOs with median abundance less than 2 × 10^−8^ in the sample were discarded. The differential KOs were identified by the Wilcoxon rank-sum test (R package wilcox.test) with a threshold *p*-value < 0.05 between the results before (T0) and after (T3) taking probiotics for three months.

The Spearman correlation coefficient (R package cor.test) was used for correlation analysis between differential KOs and differential species, between differential KOs and metabolites, and between differential KOs and clinical indexes. The confidence level was 0.95. The cutoff was the *p*-value < 0.05 (marked with a plus sign ‘+’). If the *p*-value was < 0.01, it indicated a significant correlation (marked with an asterisk ‘*’).

### 2.6. Cytokines and Biomarker Analysis

The levels of plasma interleukin 2 (IL-2), interleukin 4 (IL-4), interleukin 6 (IL-6), interleukin 10 (IL-10), interleukin 17A (IL-17A), and interferon-gamma (IFN-y) were quantitatively determined by magnetic bead-based multiplex immunofluorescence assays using the Human Th1/Th2/Th17 cytokine kit (JiangXi Cellgene, Nanchang, China) on flow cytometry (BD FASCantoll, San Jose, CA, USA), following the manufacturer’s protocol. The BD FCAP Array software (3.0.1) was used to analyze the data and output the cytokine concentrations (pg/mL).

The plasma biomarkers, occludin, zonulin, CD14, and LBP were quantified by enzyme-linked immunosorbent assay (ELISA) according to the manufacturer’s protocol. Occludin and zonulin levels were measured using ELISA kits from Cusabio (#CSB- EQ027649HU and #CSB- EL016263HU; Cusabio, China), while CD14 and LBP were measured using ELISA kits from Invitrogen (#EHCD14, #EH297RB; Invitrogen, CA, USA). The microplate reader, SpectraMax iD5 (Molecular Devices, CA, USA), was used to detect the optical density (OD) of colored reactions at a 450 nm wavelength. The output data was then analyzed using CurveExpert 1.4.0 (Hymas developers)

## 3. Results

### 3.1. Metagenomic Analysis Reveals an Altered Human Intestinal Microbiome and Augmented Beneficial Microbes with Probiotic Intervention

In order to investigate the compositional changes in intestinal microbiota following 12 weeks of probiotic intervention and another 12 weeks after discontinuity, we performed a metagenomic analysis (Appendix A in the Appendix A). The taxon differential analysis showed that 15 species, including *C. butyricum*, were enriched in the T3 data after probiotics (Wilcoxon rank-sum test, *p* < 0.05; Appendix A in the Appendix A). After considering the additional T6 data microbiota after probiotic discontinuity (Kruskal–Wallis test, *p* < 0.05; Appendix A in the Appendix A), five species closely accompanying probiotic intervention were discovered (Figure 2 and Appendix A), including *Coprobacillus* sp_D6, *Coprobacillus* sp_29_1, *Candidatus Stoquefichus_massiliensis, Carnobacterium divergens*, and *Corynebacterium massiliense*. In the control group (CG), *Peptoniphilus* sp*_oral_taxon_375*, *Lactobacillus zeae*, and *Firmicutes bacterium_ZOR0006* values varied significantly (*p* < 0.05), which reflected baseline fluctuations that were probably due to seasonal changes (Appendix A and Appendix A in the Appendix A).

Further investigating the varied abundance of beneficial symbionts, we found that genera *Bifidobacterium* and *Subdoligranulum*, and species *Akketmanse muciniphila* and *Ruminococcus obeum* (Rank-sum test, *p* < 0.05) obviously increased with the intervention process (Appendix A in the Appendix A); however, *C. butyricum* did not stably colonize after discontinuation of its use in the human intestine (Appendix A in the Appendix A).

### 3.2. Functional Annotation and Analysis Revealed Concentrated Differential KOs and Active Microbial Activities with Probiotics Compared with Scattered Changes in Control

To analyze the functional changes in the intestinal microbiome, we annotated the gut gene catalogs using the Kyoto Encyclopedia of Genes and Genomes (KEGG) database and identified 5852 KEGG orthologous groups (KOs). Wilcoxon rank-sum test (*p* < 0.05) was used to identify the differential KOs before (T0) and after (T3) taking probiotics for 12 weeks. A total of 27 differential KOs in the Test Group (TG) were identified, with 14 enriched in T3 and 13 enriched in T0 (Figure 3; Appendix A and Appendix A in the Appendix A). In particular, the increased 14 KOs included four gut microbe fermentation metabolites/enzymes (K20025, K04034, K18285, K20447), three structure components/enzymes (K02655, K00783, K09760), two sugar isomerase (K17195, K01810), and three transport proteins (K02027, K17686, K02034) (Appendix A in the Appendix A). Meanwhile, Wilcoxon rank-sum test (*p* < 0.05) in the CG showed a much-scattered differentiation with 71 increased KOs and 45 decreased KOs (Figure 3 and Appendix A, Appendix A in the Appendix A).

We further analyzed and compared the differential KOs of the two groups through the statistics of pathways in Level 1 and Level 2 KEGG (Figure 3, Appendix A in the Appendix A). In the TG, the distribution of the differential KOs was concentrated in pathways of Metabolism of Cofactors and Vitamins (14.2%) and Carbohydrate Metabolism (14.2%). In contrast, differential KOs in the CG were relatively concentrated on Membrane Transport (19.7%) and Carbohydrate Metabolism (19.7%) (Figure 3).

### 3.3. Quantitative Plasma Metabolomic Changes Involved in the Glycine and One-Carbon Metabolism with Probiotics, Compared with Bile Acid and Lipid Metabolism in the Control

In order to characterize the plasma metabolome changes after probiotic supplementation, we performed a quantitative targeted metabolomic analysis using the Q300 Kit (Metabo-Profile, Shanghai, China) and ultra-performance liquid chromatography, coupled to a tandem mass spectrometry (UPLC-MS/MS) system. In total, 204 plasma metabolites were detected and quantified. The differential analysis of the plasma metabolites between T0 and T3 by the Wilcoxon rank-sum test or Student’s *t*-test (*p* < 0.05) showed that four metabolites, i.e., glycylproline, glycolic acid, oxalic acid, and sarcosine increased in abundance, while the three radios of metabolites decreased (glycine/sarcosine, ketoleucine/leucine, and dimethylglycine/sarcosine) (Figure 2, Appendix A in the Appendix A). In the CG, the six metabolites increased (anserine, glycylproline, valeric acid, oxalic acid, GCA, and GDCA), and the EPA was reduced (Appendix A in the Appendix A). The oxalic acid and glycylproline variations were shared between the TG and CG groups, representing common basal fluctuations that were probably due to seasonal changes. The differential metabolites are discussed in more detail, later in this article.

### 3.4. Correlation Analysis between Intestinal Differential Microbes and the Plasma Metabolome Revealed Coordinated Changes in Generating SCFAs, MUFA, LUFA, and Efficient Energy Production in the TG

In order to explore the association between gut microbiota changes and the plasma metabolome, we combined metagenomics and metabolomics data and performed a correlation analysis between the different microbial species, accompanied by probiotic supplementation and all detected plasma metabolites, using the Spearman correlation coefficient (confidence level = 0.95, *p* < 0.05).

Our results revealed that 46 metabolites/ratios were correlated with the six species of *Coprobacillus* sp*_D6*, *Coprobacillus* sp*_29_1*, *Candidatus Stoquefichus_massiliensis*, *Carnobacterium divergens*, *Corynebacterium massiliense*, and *C. butyricum* (*p* < 0.05, Figure 4). Among them, 31 metabolites/ratios were positively correlated to the six species, while 15 metabolites/ratios had a negative correlation (Figure 4). Metabolites that were correlated with differential microbial species were mainly medium- to long-chain unsaturated fatty acids (MUFA and LUFA) and the different SCFA-associated carnitines, amino acids, bile acids, and metabolites involved in energy generation.

### 3.5. Significant KOs Were Correlated with Multiple Plasma Metabolites and Species

A Spearman correlation analysis was performed between differential KOs and plasma metabolites (*p* < 0.05, differential metabolites, and all detected metabolites in Appendix A in the Appendix A, respectively). The most significant correlation (*p* < 0.01) was found between K18285 (aminodeoxyfutalosine synthase, EC:2.5.1.120) and the multiple bile acid metabolites (GCDCA, TDCA, GCA, LCA), amino acids (arginine, 4-hydroxyproline), and long-chain unsaturated fatty acylcarnitine (oleylcarnitine and linoleylcarnitine). It was also significantly correlated (*p* < 0.05) with more bile acids (dihomo-gamma-linolenic acid, GDCA, GHDCA, and GUDCA) and multiple long-chain unsaturated fatty acids (gamma-linolenic acid, arachidonic acid, EPA).

Spearman correlation analysis (*p* < 0.05) was also performed between the differential KOs and differential species. Specifically, K18285, enriched at T3, was also found to be positively correlated to *C. butyricum*. K20025 (R-2-hydroxyisocaproyl-CoA dehydratase alpha subunit, EC:4.2.1.157) and K04034 (anaerobic magnesium-protoporphyrin IX monomethyl ester cyclase, EC:1.21.98.3) enriched in T3 were positively correlated to *Coprobacillus* sp*_29_1, Coprobacillus* sp*_D6,* and *Candidatus Stoquefichus massiliensis*. K01659 (2-methylcitrate synthase, EC:2.3.3.5), enriched at T0, was negatively correlated to *C. butyricum, Coprobacillus* sp*_29_1, Coprobacillus* sp*_D6* and *Candidatus Stoquefichus massiliensis* (Appendix A in the Appendix A).

### 3.6. Biomarker Assessment Indicated Improvement in Immunity and Nutrition

Six cytokines (IL-2, IL-4, IL-6, IL-10, IL-17A, and IFN-γ) that are related to inflammation and immune response regulation, and four biomarkers (occludin, zonulin, CD14, LBP) that are indicative of gut integrity were quantitatively analyzed by flow cytometry and ELISA methodologies, respectively. Differential analysis showed that most of the examined cytokines and proteins slightly decreased over time in both the TG and CG groups, reflecting a basal fluctuation trend that was probably caused by seasonal changes. Specifically, the proinflammatory factor interferon-gamma (IFN-γ) decreased in the TG (Wilcoxon rank-sum test, *p* < 0.05, Appendix A in the Appendix A), but the level was elevated in the CG. Both IL-2 and IL-10 decreased in the TG (TG, *p* < 0.05), while IL-2 also decreased in the CG (*p* < 0.05). Additionally, the tight junction protein occludin increased in the TG (*p* = 0.059); in contrast, it showed a decrement in the CG. Zonulin levels decreased in both groups (*p* = 0.009).

Three clinical nutrition biomarkers, i.e., serum albumin (AB), prealbumin (PA), and hemoglobin (Hb), were also quantitatively examined in the clinical lab of a tertiary hospital. Augmented levels of the sensitive nutrition status biomarker PA in the TG were observed (T0:221.57 ± 50.51 mg/L; T3:231.57 ± 45.86 mg/L, Table 1) in contrast to its decrement in the CG (C0: 222.83 ± 51.67 mg/L; C3: 215.33 ± 52.33 mg/L). The other two clinical parameters, AB and Hb, were slightly reduced in both groups, reflecting a shared basal fluctuation. Taking into account the relatively short intervention time, the improvement of PA as a more sensitive nutrition indicator, compared with the divergent changing trend of CG, could be able to show the potential for nutrition improvement with intervention.

## 4. Discussion

The gut microbiota of elderly people is varied and is associated with age, gender, diet, geriatric diseases, the frequent use of medicine, residential care, and socioeconomic variables. It is also more heterogeneous than in young populations [24]. Previous studies have revealed that malnutrition could affect immunity [10], while imbalanced gut microflora could affect nutrient metabolism. This imbalance could also cause “gut leakage”, which has been associated with greater susceptibility to infections and higher morbidity from chronic diseases [1,11,12]. Gut microbiota-targeted intervention, such as probiotics, has been considered a promising means of improving gut health and immunity, garnering increasing academic interest and representing a research direction showing rapid progress. Due to its efficiency and convenience for health modulation, probiotic intervention has great potential, especially for bed-dwelling elderly people. However, due to challenges in terms of sample collection, such an intervention has not yet been thoroughly studied. Herein, we focus on elderly people in long-term care in a community hospital, most of whom were bed-dwelling and recovering from a stroke, with partial loss of mobility independence, difficulty ingesting or speaking, or diminished cognitive capacity. We adopted the spore-forming, butyrate-producing *Clostridium butyricum* strain, which has been successfully used in the treatment of AAD, for microbiota improvement after surgery [25], and after cancer treatment [22]. Subjects with malnutrition according to the criteria of MNA-SF (score ≤ 7; average albumin 36.38 ± 2.98 g/L) were recruited for probiotic supplementation lasting for 12 weeks from mid-October to mid-January, and with continuous observation for another 12 weeks after probiotic discontinuation (Figure 1). All the participants or their children provided informed consent, which was followed by random assignment into a test group (TG, with probiotic) or control group (CG, no probiotic). The clinical background related to age, gender, BMI, diseases, the drugs taken, and nutrition status were comparable in both groups. The main findings can be grouped into the following aspects.

First, the metagenomic analysis revealed an altered gut microbiome with improved beneficial bacteria from the supplementation of a single strain of *C. butyricum*. The intervention did not affect the α-diversity (Shannon index and Simpson index) of the gut microbiota. Based on the rank-sum test (*p* < 0.05) and KW analysis (*p* < 0.05), five differential bacteria species accompanying probiotic usage were identified. Among them, two *Coprobacillus* species, which may be involved in energy absorption, were related to obesity and diabetes [26]. They were also enriched in longevous populations [27]. *Carnobacterium divergens* is a lactic acid-producing bacterium (LAB) that could reside in skin and mucus and could compete with other bacteria via the secretion of bacteriocin acid, thus helping to maintain barrier integrity [28]. *Candidatus Stoquefichus massiliensis* and *Corynebacterium massiliens* are usually considered to be commensal bacteria. Several *Clostridium* species and beneficial symbionts such as *Alistipes putredinis* also grew in abundance (rank-sum test, *p* < 0.05), with the latter conferring the ability of indole and SCFA production for inflammation-associated mental disorder regulation [29]. *Akketmanse muciniphila* and *Ruminococcus obeum*, as well as *Bifidobacterium* and *Subdoligranulum* genera microorganisms, also notably increased in abundance; however, this was without statistical significance (Appendix A in the Appendix A). In the CG, differentially varied *Firmicutes bacterium* ZOR0006, *Peptoniphilus* sp. oral taxon 375, and *Lactobacillus zeae* (*p* < 0.05), illustrated a basal-level microbiota fluctuation along with seasonal change.

The functional annotation against the Kyoto Encyclopedia of Genes and Genomes (KEGG) database and analysis of metagenomics showed a concentrated shift in active microbial functions with probiotic supplementation and a scattered change in the CG. In the TG, 5852 KEGG orthologs (KOs) were annotated, with 14 increased and 13 decreased KOs (*p* < 0.05, Appendix A in the Appendix A). Comparatively greater variation (71 KOs increased; 45 KOs decreased) was observed in the CG. The differential TG KOs contained gut microbial fermentation enzymes, structure components, sugar isomerases, and transport proteins, which were enriched in cofactor/vitamin metabolism, and carbohydrate metabolism pathways, demonstrating an activated microbial fermentation and function. In contrast, diversely distributed differential KOs in the CG were enriched in carbohydrate metabolism and membrane transport. Both groups’ shared carbon metabolism elevations indicated efficient energy generation in winter [30]. Active membrane transporting has also previously been reported; however, this was only in hibernating animals [31]. Age-related vitamin metabolism is a hallmark of health that is related to vegetable intake [11].

Second, plasma metabolomic studies showed that in contrast to bile acid and lipid metabolism in the CG, probiotics likely activated the glycine and one-carbon metabolism involved in cell proliferation potential and active energy generation in the TG. The metabolites glycylproline, glycolic acid, oxalic acid, and sarcosine increased significantly (*p* < 0.05), and the ratios of glycine/sarcosine, ketoleucine/leucine, and dimethylglycine/sarcosine decreased in the TG. All the increased metabolites are involved in glycine metabolism, which is a key component in one-carbon metabolism, with potential relevance to the TCA cycle and cell proliferation regulation [32]. In the CG, six metabolites rose abundantly (anserine, glycylproline, valeric acid, oxalic acid, GCA, and GDCA), and EPA decreased. Glycylproline and oxalic acid increased in both groups, and oxalic acid could be endogenously produced by *Clostridium, Bifidobacteria*, or *Lactobacillus* [33] for the formation of uracil and orotic acid (Vitamin 13) [34]. The CG-enriched metabolites, GCA, GDCA, and anserine, are involved in bile acid metabolism and could reflect lipid and cholesterol metabolism, while the TG could maintain bile acid homeostasis. GDCA has been reported to be related to impaired insulin clearance which induced obesity-associated hyperinsulinemia [35]. The increase in GDCA is also related to Alzheimer’s disease [36]. The increase in GCA is related to chronic liver disease [37]. Valeric acid could be considered a biomarker for liver fat and a predictor of diabetes [38].

Third, the correlation between significantly varied bacteria (SCFA-producing *C. butyrate*, energy-boosting *Coprobacillus*, amino acid-catalytic *C. divergens*, and *Coprobacillus*) and metabolism suggested coordination in generating SCFAs, MUFA, and LUFA, and efficient energy production in the TG (Figure 4). Most of the increased bacteria were positively related (*p* < 0.01) to metabolites, including unsaturated medium- to long-chain fatty acids (MUFA, LUFA) (gamma-linolenic acid, myristoleic acid, docosapentaenoic acid), carnitines, and SCFA derivatives (L−carnitine, propionyl carnitine, acetylcarnitine, and valerylcarnitine), and multiple amino acids (citrulline/arginine, asparagine, and ornithine) (Figure 4). Among them, MUFA and LUFA are hallmarks of healthy cardiovascular and neuroprotection activity. SCFA benefits gut health from multiple aspects. Citrate acid and its precursor, cis−aconitic acid, are involved in the TCA cycle for carbohydrate metabolism and efficient energy production. Carnitines are involved in lipid metabolism and energy production by mitochondrial oxidation. Comparatively speaking, correlations in the control group were rather scattered. Furthermore, *Akketmanse muciniphila* is not only involved in short-chain fatty acid metabolism but also in bile acid metabolism [39,40].

Correlation analysis was also performed between significantly varied KOs and all detected metabolites (Appendix A in the Appendix A). A distinctive correlation (*p* < 0.05) was observed between K18285 (aminodeoxyfutalosine, AFL synthase, EC:2.5.1.120), and over 15 metabolites, including LUFA and LUFA-carnitines, bile acids, and amino acids. AFL synthase is a radical SAM enzyme encoded by the microbial gene, *mqnE*, known as an anti-inflammatory factor and a key intermediate factor in the production of the essential vitamin K2. Moreover, it is critical in maintaining coagulation homeostasis and the prevention of osteoporosis, and is also associated with improvement in Parkinson’s disease [41]. Of note, it was also found to be significantly correlated with add-in strain *C. butyricum* (*p* < 0.05, Appendix A in the Appendix A). Thus, the adoption of *C. butyricum* could boost the activity of AFL synthase and be involved in the general metabolism activation of bile acids, amino acids, and LUFA-carnitines. Another significant correlation (*p* < 0.05) was between K02655 (type-IV pilus assembly protein PilE) and multiple SCFAs (propionic acid, butyric acid, and isovaleric acid), amino acids (aspartic acid/beta-alanine, tyrosine, glycylproline, and creatine), bile acids, and organic acids. Type-IV pili are ubiquitous (30–45% of gut microbes) in bacteria and have an important interface with the host [42]. Correlation analysis in this study suggested that the actively proliferating bacteria (with higher pili density) in the TG could greatly impact the metabolome.

Moreover, plasma cytokines (IL2, 4, 6, 10, 17A, and IFN-γ), gut integrity markers (occludin, zonulin, CD14, and LBP), and nutrition markers (PA, AB, and Hb) were quantitatively examined. Most of the examined cytokines and proteins slightly varied in both groups, implying a lowered immunity along season change [43,44]. Proinflammatory factor IFN-γ decreased in the TG (*p* < 0.05), in contrast with an insignificant increment in the CG. Both IL-2 and IL-10 were decreased in the TG (*p* < 0.05); IL-2 also decreased in the CG (*p* < 0.05). Tight junction protein occludin increased in the TG and decreased in the CG (*p* = 0.059). However, zonulin decreased in both groups, which may be related to previously reported cases of improved defecation [45]. Thus, the distinctive changes in IFN-γ and occludin showed the possibility of immunity improvement with probiotic intervention. The sensitive nutrition biomarker PA showed an increasing trend that was divergent from its decrement in the CG, indicating a potential for nutrition improvement (Table 1).

The clinical records related to the diagnosis of diseases and drugs, along with questionnaires on defecation state and frequencies, neuropsychic activities, and subjects’ appetite at the beginning, middle, and end of the intervention period; the changes were analyzed (Appendix A in the Appendix A). During the intervention, the rate of exclusion (caused by the onset of diseases, the use of antibiotics and other probiotics including yogurt), diet stability, drugs, and the treatment of chronic diseases were comparable between the two groups. In addition to improved defecation tenderness or frequency (7/11 TG persons), more active neuropsychic activities in terms of speaking ability, movement, mental state, or the recognition of persons or things were also observed (5/11 persons). Given the relatively short intervention time and the improved immunity and nutrition status, which diverged from the CG, the probiotic showed promising potential. However, the relatively high dropout ratio in both TG and CG restricted the number of qualified subjects and samples for subsequent analysis in this work. Thus, a further study with a longer intervention, a larger sample size, and a more comprehensive analysis is needed to thoroughly investigate the global impact of probiotics on the health of elderly people with malnutrition. Gnotobiotic animal or enterocyte studies may be needed to find the molecular mechanism underlying this effect.

Subgroup analysis using a multivariate association with the linear model (MaAsLin) between the clinical phenotypes and microbiome/metabolism of this study showed that the differential species across diseases (CAD, HBP, DM), after adjusting for age, BMI, and sex were *Actinomyces graevenitzii, Paracoccus sanguinis,* and *Enhydrobacter aerosaccus* (FDR < 0.1). The metabolites glutamic acid/glutamine and aspartic acid were negatively correlated with age, and 2-furoic acid was positively correlated (FDR < 0.1, Appendix A in the Appendix A). Fumaric acid/malic acid, tetradecanoylcarnitine, and malonylcarnitine were differential and were higher in males (FDR < 0.1). These substrates were not substantially changed in the metagenomics or metabolomics data analyzed and reported in the last part of this study. Further investigations could adopt a larger sample size for better hierarchy analysis. To gain a broader knowledge for positioning this study’s metagenomes with former research, we further compared our data to the latest available metagenome databases from 7 cohorts, including IBD patients [46], RA patients [47], metabolic syndrome patients [48], fatty liver patients [49] T2D patients [50], and longevous participants (94–105 years old) and their offspring (50–79 years old) [27]. Interestingly, through a comparison with 7 cohorts, we found that the gut microbiota structure of our data was closest to the longevous people and furthest from the diseased groups (Appendix A in the Appendix A). We also found that the elevated taxa (*Coprobacillus*) and KOs (K09760, DNA recombination protein RmuC; K17195, D-allulose-6-phosphate 3-epimerase) were enriched in the longevous group and in our cohort [27].

Briefly, probiotic intervention conducted over a seasonal change has shifted the gut microbiome and host metabolome. A comparatively stable and functionally concentrated differential gut microbiota has been described in this study, which correlated well with the impacted metabolomes that are especially involved in activated SCFA production, bile acid metabolism, and energy generation. In addition, boosted growth was observed in *Coprobacillus* and the beneficial microbes *Akketmanse muciniphila, Alistipes putredinis,* and *Subdoligranulum*, which are related to health and longevity [27]. With the seasonal change, the immunity and nutrition indicators generally decreased slightly in both groups, but after a probiotic intervention, a divergent and significant change in IFN-γ, occludin, and PA levels could indicate a promising improvement potential. To sum up, this study provides evidence that promotes our understanding of the effect of probiotics on multiple aspects of the health of older adults suffering from malnutrition. These data will also further the research of health modulation in elderly populations.

## 5. Conclusions

In summary, this 12-week, randomized, single-blind clinical trial used quantitative metagenomics, metabolomics, biomarkers, and clinical analysis to examine the effect of probiotic Clostridium butyricum on elderly people with malnutrition in long-term care (MNA-SF score of ≤ 7). Our results revealed altered gut microbiomes with the promoted growth of beneficial microbes, such as *Akketmanse muciniphila*, and changes in functional orthologs enriched in cofactor/vitamin production and carbohydrate metabolism pathways, which were positively correlated with elevated plasma metabolites SCFA, MFA, LFA, carnitines, and amino acids that are indicative of coordinated and ameliorated metabolism with probiotic intervention. Considering the relatively short intervention time and the improved immunity and levels of nutrition biomarkers such as IFN-γ, occludin, and prealbumin, *C. butyricum* could improve immunity and nutrition in older adults with malnutrition. 

## Figures and Tables

**Figure 1 nutrients-14-03546-f001:**
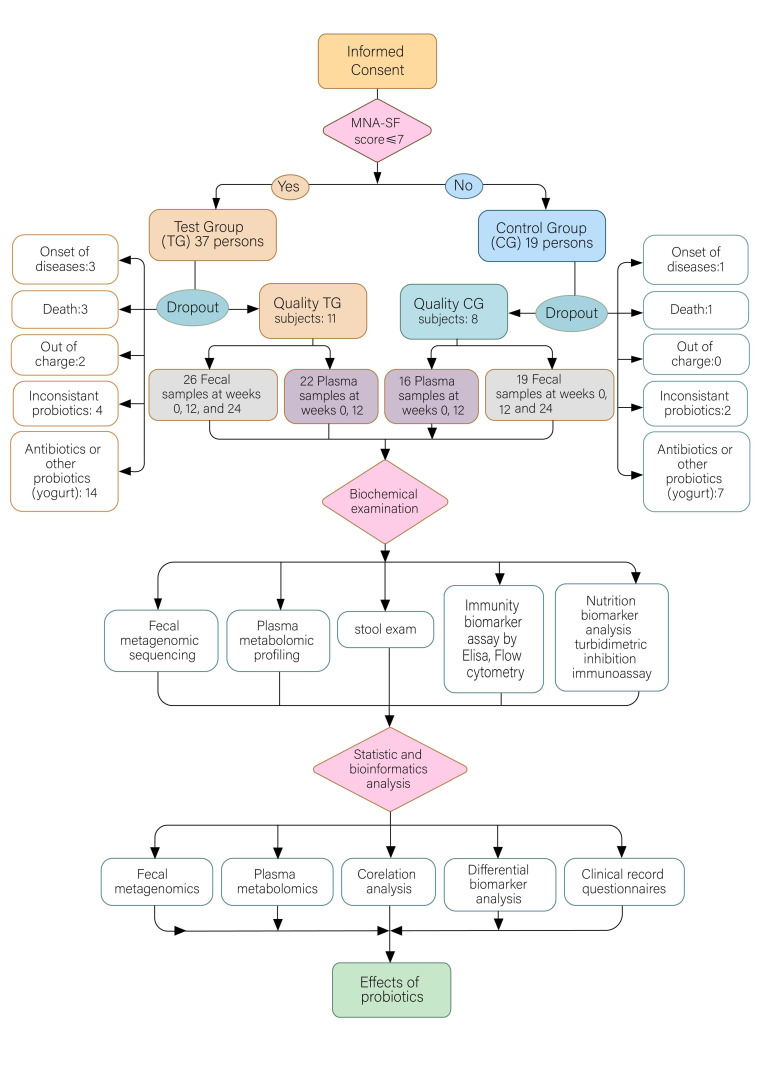
Flowchart of the probiotic intervention study with enrolled elderly subjects.

**Figure 2 nutrients-14-03546-f002:**
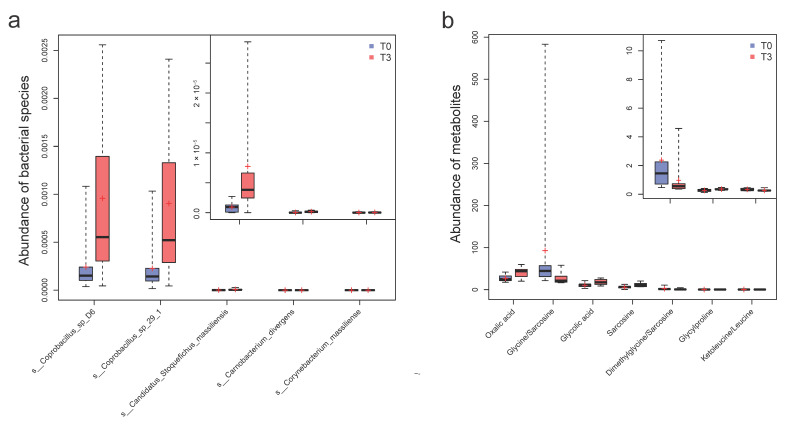
The significantly varied species and metabolites accompanying probiotic supplementation. (**a**) The five differential species throughout the intervention (T3 vs. T0) identified by the Wilcoxon rank-sum test (*p* < 0.05) and Kruskal–Wallis test (*p* < 0.05), with the incorporation of T6 samples after probiotic discontinuity. (**b**) The seven differential metabolites (T3 vs. T0) identified by Wilcoxon rank-sum test (*p* < 0.05). Samples in the T3 group are shown in red, and samples in the T0 group are shown in blue.

**Figure 3 nutrients-14-03546-f003:**
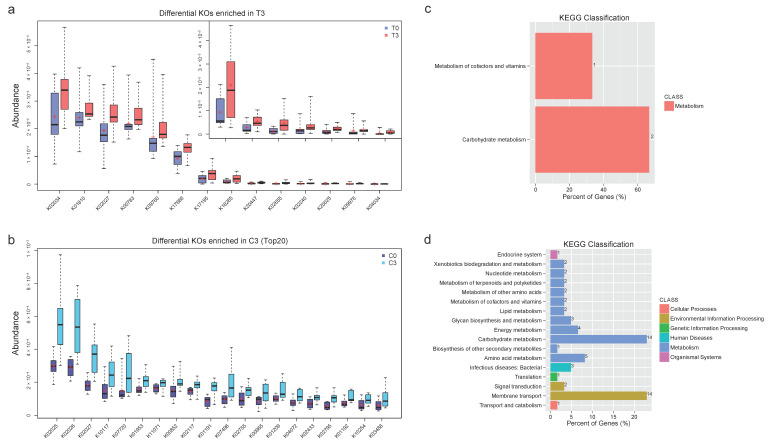
Functional analysis of differential KOs and the enrichment of the KEGG pathway. (**a**). The differential KOs were identified with the Wilcoxon rank-sum test (*p* < 0.05) between T0 and T3 groups. (**b**) Significantly differential KOs (*p* < 0.05) in the control group. (**c**) The KEGG classification of the differential KOs in the test group with the Level 1 and Level 2 information; (**d**). The KEGG classification of the differential KOs in the control group.

**Figure 4 nutrients-14-03546-f004:**
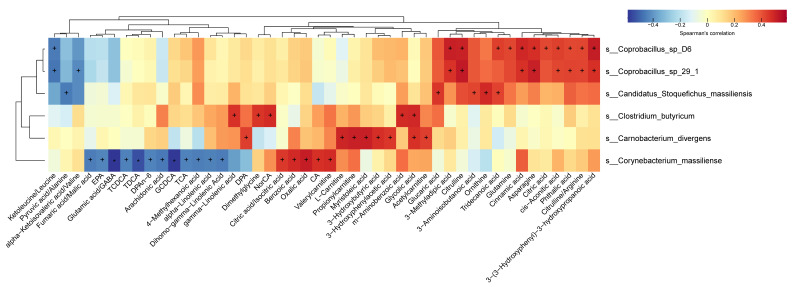
Correlation analysis of the differential species and all detected plasma metabolites. The correlation analyses between the differential species within the intervention period and all the detected plasma metabolites were conducted by the Spearman correlation coefficient. Color depth indicates the strength of correlation, with red representing positive correlation and blue for the negative correlation. An asterisk ‘+’ indicates *p* < 0.05, and an asterisk ‘*’ stands for *p* < 0.01.

**Table 1 nutrients-14-03546-t001:** Clinical characteristics of the participants.

Clinical Characteristics	Probiotic Group (*n* = 11)	Control Group (*n* = 8)
Age (year), mean ± SD	81.64 ± 5.01	85.38 ± 4.92
Gender (*n*), female/male	8/3	5/3
BMI (kg/m^2^), mean ± SD	19.64 ± 5.29	21.18 ± 1.64
AB (g/L), mean ± SD	37.14 ± 2.53 (0 weeks)/36.29 ± 3.61(12 weeks)	37.50 ± 1.80 (0 weeks)/34.50 ± 2.63 (12 weeks)
PA (mg/L), mean ± SD	221.57 ± 50.51 (0 weeks)/231.57 ± 45.86 (12 weeks)	222.83 ± 51.67 (0 weeks)/215.33 ± 52.33 (12 weeks)
Hb (g/L), mean ± SD	107.71 ± 11.57 (0 weeks)/102.14 ± 21.56 (12 weeks)	126.33 ± 21.08 (0 weeks)/115.33 ± 24.57 (12 weeks)
CI disease, *n* (%)	9 (81.82)	8 (100.00)
HBP disease, *n* (%)	10 (90.91)	7 (87.50)
CAD disease, *n* (%)	9 (81.82)	6 (75.00)

BMI, body mass index; AB, albumin; PA, prealbumin; Hb, hemoglobin; CI, cerebral infarct; HBP, high blood pressure; CAD, coronary heart disease.

## Data Availability

The Illumina raw-read data have been deposited at the National Center for Biotechnology Information (NCBI) under accession number PRJNA827603. A STORMS (Strengthening the Organizing and Reporting of Microbiome Studies) checklist is available at https://doi.org/10.5281/zenodo.6497355 (accessed on 27 April 2022).

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
