# Peer review of "Clostridium butyricum Potentially Improves Immunity and Nutrition through Alteration of the Microbiota and Metabolism of Elderly People with Malnutrition in Long-Term Care"

_nutrients, 2022, doi:10.3390/nu14173546_

Round 1

Reviewer 1 Report

My main comment is that cohort they used in the study is very small to make any impactful conclusion and this should be stated and this study treated as a pilot study for a larger longitudinal study. Also the timing pre-treatment and post-treatment seems very shorts.

Finally few enzymes and potential metabolic pathway were highlighted in the metabolic study , however the authors focus on short and long chain fatty acids that are metabolites widely studied and already connected to anti-inflammatory role in individuals at any age. This really affect the originality of the study. Another metabolite/pathway could be extrapolated from these data? If not is this study really adding anymore to our knowledge in the field?

More data from the metabolomics should be presented or explained to justify the importance pf the bacterial species defined as beneficial in the elderly hosst.

Reviewer 2 Report

It is an interesting work in itself but difficult to understand. Authors must review the form in English, even with the help of a native speaker. They must also clearly indicate the statistical methodology and the software package used

Author Response

Response to Reviewer 2 Comments

Point 1: It is an interesting work in itself but difficult to understand. Authors must review the form in English, even with the help of a native speaker.

Response 1: Thank you for your comments on the manuscript. According to your advice, I have found a professional English Language Editing service and had the manuscript checked and polished. Please find the attached English Language Editing Certificate. 

Point 2: They must also clearly indicate the statistical methodology and the software package used.

Response 2:  Thanks a lot for your advice. We have added more details to Introduction, Materials and Methods, including the information of the databases and software used.

A new reference was added (Gingrich, A., et al., Prevalence and overlap of sarcopenia, frailty, cachexia and malnutrition in older medical inpatients. BMC Geriatrics, 2019. 19(1): 120) in Line 59 following the sentence “Malnutrition, frailty, sarcopenia, and immunosenescence are typical health concerns with high incidence among elderly populations.”

Line 167: The full name of KEGG database was added as “the Kyoto Encyclopedia of Genes and Genomes)”

Line 171: The NCBI database information was added as “http://www.ncbi.nlm.nih.gov”.

Line 172: The version of software SOAPalign was added as “version 2.21”.

Line 202: The software information was added as “Wilcoxon rank-sum test (R package wilcox.test)”

Line 209: The software information was added as “Kruskal-Wallis test (R package kruskal.test)”.

Line 217: The software information was added as “Spearman correlation coefficient (R package cor.test)”.

Huanlong Qin, PhD

Institute of Intestinal Diseases, Shanghai Tenth People’s Hospital, Tongji University School of Medicine, Shanghai, China

Tel: +86 18001763088

Email: qinhuanlong@tongji.edu.cn

Lin Liu, PhD

Institute of Intestinal Diseases, Shanghai Tenth People’s Hospital, Tongji University School of Medicine, Shanghai, China

Tel: +86 15157111890

Emai: lindaliu79@163.com

August 23, 2022

Response to Reviewer 2 Comments

Point 1: It is an interesting work in itself but difficult to understand. Authors must review the form in English, even with the help of a native speaker.

Response 1: Thank you for your comments on the manuscript. According to your advice, I have found a professional English Language Editing service and had the manuscript checked and polished. Please find the attached English Language Editing Certificate. 

Point 2: They must also clearly indicate the statistical methodology and the software package used.

Response 2:  Thanks a lot for your advice. We have added more details to Introduction, Materials and Methods, including the information of the databases and software used.

A new reference was added (Gingrich, A., et al., Prevalence and overlap of sarcopenia, frailty, cachexia and malnutrition in older medical inpatients. BMC Geriatrics, 2019. 19(1): 120) in Line 59 following the sentence “Malnutrition, frailty, sarcopenia, and immunosenescence are typical health concerns with high incidence among elderly populations.”

Line 167: The full name of KEGG database was added as “the Kyoto Encyclopedia of Genes and Genomes)”

Line 171: The NCBI database information was added as “http://www.ncbi.nlm.nih.gov”.

Line 172: The version of software SOAPalign was added as “version 2.21”.

Line 202: The software information was added as “Wilcoxon rank-sum test (R package wilcox.test)”

Line 209: The software information was added as “Kruskal-Wallis test (R package kruskal.test)”.

Line 217: The software information was added as “Spearman correlation coefficient (R package cor.test)”.
